# Adsorption Studies of Waterborne Trihalomethanes Using Modified Polysaccharide Adsorbents [note 1]

**DOI:** 10.3390/molecules26051431

**Published:** 2021-03-06

**Authors:** Rui Guo, Lalita Bharadwaj, Lee D. Wilson

**Affiliations:** 1Department of Chemistry, University of Saskatchewan, 110 Science Place, Saskatoon, SK S7N 0X2, Canada; rug194@mail.usask.ca; 2Department of Community Health and Epidemiology, University of Saskatchewan, Health Sciences Building, 107 Wiggins Road, Saskatoon, SK S7N 5E5, Canada; lalita.bharadwaj@usask.ca

**Keywords:** trihalomethanes, adsorption isotherms, β-cyclodextrin, chitosan, polymers, fractionation

## Abstract

The adsorptive removal of trihalomethanes (THMs) from spiked water samples was evaluated with a series of modified polysaccharide adsorbents that contain β-cylodextrin or chitosan. The uptake properties of these biodegradable polymer adsorbents were evaluated with a mixture of THMs in aqueous solution. Gas chromatography employing a direct aqueous injection (DAI) method with electrolytic conductivity detection enabled quantification of THMs in water at 295 K and at pH 6.5. The adsorption isotherms for the polymer-THMs was evaluated using the Sips model, where the monolayer adsorption capacities ranged between 0.04 and 1.07 mmol THMs/g for respective component THMs. Unique adsorption characteristics were observed that vary according to the polymer structure, composition, and surface chemical properties. The modified polysaccharide adsorbents display variable molecular recognition and selectivity toward component THMs in the mixed systems according to the molecular size and polarizability of the adsorbates.

## 1. Introduction

Maintaining drinking water quality is a significant global issue in public health, where the use of chlorine-based disinfectants (e.g., Cl_2_ or sodium hypochlorite) in drinking water is regarded as a major achievement for public health in the 20th century [1]. Chlorination efficiently reduces microbial pathogen levels in water supplies and the incidence of waterborne diseases. An unintended consequence of the disinfection process results in the formation of disinfection by-products (DBPs) for water that contains dissolved organic carbon (DOC) [2,3]. Trihalomethanes (THMs) are among the most common and important DBPs, which pose a potential public health concern as a result of conventional water disinfection processes. THMs are halomethanes with the general molecular formula CHX_3_, where X represents fluorine, chlorine, bromine, or iodine (cf. Figure 1; inset). Several important THMs present in disinfected water supplies are chloroform, bromodichloromethane (BDCM), dibromochloromethane (DBCM), and bromoform (cf. Table 1 & Table 2). The formation of THMs is closely related to the specific conditions in the water environment such as chlorine dosage, contact time, pH, bromide ion concentration, and levels of DOC in the source water supply [2].

THMs are highly volatile organic compounds, where their physical properties are listed in Table 1. Chloroform is volatile and is partially soluble in water with greater solubility in organic phases. Both chloroform and brominated THMs remain in air and water, with relatively long residence times, which can partition to groundwater through soil [4]. Hydrolysis of brominated THMs in aqueous media is very slow where the estimated half-lives are as follows: BDCM (1000 y), DBCM (274 y), and CHBr_3_ (686 y) [4,5].

The major routes of human exposure to THMs are ingestion, inhalation, and dermal contact with chlorinated water. Similarly, THMs are often ingested by direct contact with contaminated water and through consumption of foods containing THMs [7]. The International Agency for Research on Cancer (IARC) has classified chloroform and BDCM as possible carcinogens for humans (Group 2B), and there is sufficient evidence in the case of animal studies (cf. Table 2). DBCM and bromoform were assigned to Group 3 Environmental Protection Agency (EPA) cancer potency factors because of their inconclusive carcinogenicity in humans and limited evidence from experimental studies on animals [1,8]. Besides carcinogenicity, animal studies indicate that chloroform in drinking water supplies may lead to low birth weight, prematurity, and intrauterine growth retardation [1,8,9]. Chloroform may cause acute and chronic ecological effects to aquatic life due to its moderate acute and chronic toxicity to aquatic organisms.

The United States (U.S.) EPA has published the Disinfection By-products Rule to regulate total THMs at a maximum allowable annual average level of 80 ppb for Stage 1 and 40 ppb in Stage 2 for water treatment processes [9]. The maximum acceptable concentration (MAC) for THMs in drinking water established by Health Canada is 100 ppb. The MAC is based on a locational annual average for a minimum quarterly sampling taken at points in the distribution system with the highest potential THM levels. Thus, the removal of THMs to levels below their regulatory values, along with development of related analytical methods is an important goal toward addressing drinking water quality [10].

Adsorption-based removal of THMs represents a relatively inexpensive and facile technology for tertiary water treatment [11]. In comparison to reverse osmosis and distillation, membrane filtration is often limited to the type or size of contaminants, whereas adsorptive removal is a more versatile technology that relies on physical or chemical interactions with the adsorbent [12]. Whereas activated carbon (AC) is a common industrial adsorbent for the removal of THMs, there are potential limitations due to its production and regeneration costs due to its energy footprint [13]. AC displays variable physicochemical properties depending on its mode of preparation and functionalization [11,14,15]. By comparison, there is interest in the development of sustainable and low-cost biomaterials with tunable adsorption properties that offer an improvement in the thermodynamic and kinetic properties of adsorption for such systems [11,16].

In particular, synthetically modified polymers prepared by functionalization, cross-linking, and composite formation have been prepared to yield adsorbents with tunable physicochemical properties [17,18,19]. Synthetic reactions under thermodynamic and kinetic control that employ variable stirring rates, cross-linker ratios, and type of polysaccharide are known to yield adsorbents with controlled surface area, chemical functionality, and biodegradability [17,18,19,20,21]. Examples of such polysaccharide adsorbents that contain β-cyclodextrin (β-CD) or chitosan (CS) that can be conjugated with polyfunctional agents yield cross-linked and grafted materials. In this study, β-CD was cross-linked with sebacoyl chloride (SCl) or terephthaloyl chloride (TCl), whereas chitosan was grafted with polyacrylic acid (PAA) and β-CD. The use of variable reactant ratios at variable reaction conditions can afford materials with variable structure, composition, and physicochemical properties. β-CD is a toroidal-shaped macrocycle with amphiphilic properties with a torus length of 7.9 Å and an internal diameter between 7.5 to 8.3 Å. The toroidal cavity of β-CD lends to its unique inclusion properties, especially toward suitably sized lipophilic guest molecules (cf. Scheme 1a) [22]. The properties of β-CD have been extended to polymeric forms via cross-linking, grafting, or non-covalent self-assembly with suitable reagents to yield framework materials with unique inclusion and solid-phase extraction (SPE) properties [23,24]. CS is a linear polysaccharide composed of randomly distributed β-(1 → 4)-linked D-glucosamine and N-acetyl-D-glucosamine units that can be synthetically modified due to its abundant amine and hydroxyl groups [25,26]. Ionic and covalently cross-linked CS frameworks have been reported with adsorption sites that possess variable hydrophile-lipophile balance (HLB), in accordance with the nature of the cross-linker. In turn, such CS frameworks can be used for the controlled uptake of adsorbates due to their variable HLB character. Mohamed et al. reported on the fractionation of complex mixtures of organic acids by systematically cross-linking CS at variable glutaraldehyde content, as revealed by a scalable imprinting factor [27].

Herein, CS and β-CD (cf. Scheme 1) were used for the design of tailored SPE materials because of their tunable host–guest chemistry upon synthetic modification. The use of such versatile polysaccharides are anticipated to possess unique adsorption properties according to a facile cross-linking strategy [28]. This study reports on the uptake of THMs from water at equilibrium conditions using a series of synthetically modified polysaccharide-based adsorbents. While the synthesis and characterization of these adsorbent materials have been reported previously, the dye adsorption results for such polysaccharide (PS) materials (cf. Scheme 2) provide evidence of their variable surface chemistry and textural properties. The limited availability of adsorption studies for such polymers for the controlled removal of THMs and chlorinated organics from mixtures [11,28,29] provided the motivation for this study. Thus, the present study describes the solid–liquid adsorption properties of several cross-linked PS materials (cf. Scheme 2) with THMs in aqueous media [17,18,19].

## 2. Results and Discussion

Several types of polysaccharide materials were prepared at variable composition, where the copolymers of interest for this study are outlined in Table 3. The copolymer materials contain a polysaccharide component (β-CD or chitosan) that is cross-linked to a second component, which has been described previously, along with the structure and physicochemical characterization results elsewhere [17,18,19]. Table 3 outlines the composition of the copolymers, where it is noted that the materials were prepared as insoluble products with variable hydrophile-lipophile balance (HLB). In turn, the solid-phase adsorbent materials outlined in Table 3 allow for phase separation of the copolymers from water after the adsorption process. This enables determination of the residual (unbound) concentration (C_e_) and adsorbent amount (Q_e_) of the THMs at equilibrium after the adsorption process, according to Equation (2).

The concentration of unbound THMs (C_e_) after the adsorption process can be measured directly using the direct aqueous injection (DAI) method with gas chromatography (GC) and electron-capture detection (ECD) [30,31]. The DAI-GC-ECD method provides quantitative detection of halomethanes in water at equilibrium with the use of gas-tight vials (cf. Experimental Methods, Section 4), where the utility of DAI-GC as an analytical method for the detection of THMs was previously reported [10,31]. The adsorption properties of the copolymers with the THMs was posited to vary in accordance with the unique structure and composition of the adsorbents (cf. Table 3 and Scheme 2) [17,18,19].

Figure 1 shows that the GC-ECD chromatogram yielded good separation of THMs and the internal standard according to the resolved retention times of the component fractions. Chlorobenzene (ClBz) exhibits suitable retention behavior as an internal standard because of its partial solubility in water (0.05 g/mL), and ClBz is well resolved among the four THMs. In a previous study [31] of chloroform in aqueous solutions, the GC-based DAI method provided reliable results with adequate detection limits in the ppm region. Various types of synthetic and PS-based adsorbent materials have been reported [32,33] for the separation and capture of volatile organic carbons, in line with the methodology reported herein for the quantitative analysis of THMs. Figure 2 illustrates linear calibration curves (R^2^ > 0.99) for THMs at variable concentrations (5–50 ppm) against an internal standard solution (cf. Table 3). In turn, the adsorption capacity of the PS-based adsorbents was achieved by in situ quantitative analysis of the adsorption profiles of the THMs outlined in Figure 3.

The use of the GC-based DAI method allows for the quantitative analysis of mixtures of THMs, before and after the adsorption process. The results in Figure 3A–H illustrate the adsorption isotherm profiles (Q_e_ versus C_e_) for the polysaccharide materials, along with activated carbon for comparison. The plots in Figure 3 illustrate the adsorbed amount of THMs per gram of adsorbent as denoted by Q_e_ (mmol/g), whereas C_e_ denotes the residual concentration (mM) of THMs after the adsorption process. The adsorption properties of the PS-based materials that showed favorable chloroform adsorption (cf. Table 3) were selected for further adsorption studies with the mixture of THMs in aqueous solution. In general, the magnitude of Q_e_ increased monotonically as C_e_ increased, where differences in the relative adsorption affinity among the various adsorbents with the component THMs were observed, as described below.

### 2.1. Chitosan-Based Copolymers

In Figure 3, panels A and B show the trends in THMs uptake for the CP-1 and CP-5 adsorbents, where CP-1 showed greater overall uptake versus CP-5. In the case of CP-1, bromoform showed the greatest uptake among the THMs, whereas CP-5 had the highest uptake with chloroform. The Q_m_ (mmol/g) values for the THMs with CP-1 are given in descending order: CHBr_3_ (0.440) > DBCM (0.415) > BDCM (0.387) > CHCl_3_ (0.0420). In contrast, the Q_m_ values for CP-5 adopted a trend with a reverse order: CHCl_3_ (0.141) > BDCM (0.119) > DBCM (0.0684) > CHBr_3_ (0.0522). Considering the relative binding affinity between the PS materials with the various THM components, CP-1 had favorable adsorption with the brominated THMs that concurs with the greater apolar nature of this adsorbent due to its greater chitosan content (cf. Table 3). The importance of solvent effects and van der Waals interactions are evident from the trends in adsorption between these chitosan polymers and the trend in water solubility of the THMs (cf. Table 1). The greater chitosan content of CP-1 concurs with its greater apolar character and favorable uptake of bromoform, in line with the role of hydrophobic effects [17,31] and the trend in K_H_ values listed in Table 1 for the THMs. The greater uptake of chloroform by CP-5 concurs with the greater hydrophilic character of the adsorbent on account of its higher PAA content, along with the greater dipolar character and the water solubility of CHCl_3_ among the various THMs (cf. Table 1). Surface accessibility of the active adsorption sites may relate to the role of the hydration effects of copolymers in their swollen state, as evidenced by the greater swelling ratio (r) in water for CP-1 (r = 22.6) over CP-5 (r = 11.6) [19]. The greater hydrophilic character of CP-5 is attributed to its greater PAA content, which favors hydration of the adsorbent and the potential role of dipolar interactions with THMs that possess greater dipolar character such as CHCl_3_. CP-1 shows an incomplete saturation profile of the isotherm for CHBr_3_ over the range of experimental C_e_ values studied. This apparent effect relates to its greater adsorption affinity (cf. Table 4) and the equimolar concentration of THMs employed in the experimental design for this study were intended to achieve good accuracy and suitable dilution factors for the GC-ECD analysis. Notwithstanding the saturation limits of the isotherms, the trend in the adsorption profiles for Figure 3A,B provide a meaningful relative comparison for the uptake of THMs by the adsorbents. Table 5 lists the calculated dipole moments of the component THMs according to estimates obtained using Spartan’08 1.2.0. The parameters in Table 5 for the THMs provide complementary insight on the trend for the Q_m_ values (cf. Table 3) in these CS/PAA copolymers. The role of hydrophobic effects and the availability of polar adsorption sites (–OH, –NHR, –COO^−^), on the polymer surface play a key role, where the molecular size and polarity of the THMs exert an influence their uptake properties.

Previous isotherm studies [17,18,19] of polymer-dye systems revealed the utility of the Sips model [34]. The Sips model provides an account of adsorption profiles that are otherwise accounted for by the Langmuir and Freundlich models due to its dual model nature. Additionally, the Sips model provides an estimate of the monolayer adsorption capacity (Q_m_) and an account of the role of heterogeneous adsorption sites, as the exponential term (n_s_) deviates from unity (n_s_ ≠ 1) for such types of adsorbent systems. The adsorption profiles for the THMs were evaluated by applying the Sips isotherm (cf. Equation (4) in Section 4), where the “best-fit” results are listed in Table 4. The general utility of the Sips model and the isotherm parameters is evidenced by the goodness-of-fit to the isotherm results shown in Figure 3.

### 2.2. β-CD-Based Copolymers

The trend in the adsorption properties of the copolymer materials relates to the nature of the cross-linker, polysaccharide, and its relative composition (cf. Table 3 and Table 4). The adsorbent structure (cf. Scheme 2) can be varied accordingly by these parameters, in accordance to the synthetic conditions (e.g., reagent mole ratios and mixing speeds) as evidenced in Figure 3C–G. The THMs reported herein have variable molecular structure (cf. Figure 1, inset) and physical properties (cf. Table 1), where the Henry’s law constants of CHCl_3_ and CHBr_3_ vary over two orders of magnitude. The offset in water solubility of the THMs and their adsorption properties relate to the size, polarity, and molecular polarizability of the halomethanes, as revealed by the variable isotherms for adsorbents (cf. Figure 3C–G) that contain β-CD.

According to Figure 3, panel C reveals the highest uptake of THMs overall. The β-CD/PAA (1:5) system revealed systematically higher adsorption properties with the various THMs versus the β-CD/PAA (1:5 high speed and 1:10) systems. This may relate to the presence of its suitably sized lipophilic inclusion sites (cf. Table 6) that play a variable role according to the level of cross-linking and inclusion site accessibility. The 1:10 copolymer has attenuated adsorption capacity due to the greater PAA content or extensive cross-linking effects [17]. Further support of reduced inclusion sites accessibility of β-CD is shown for the β-CD/PAA 1:5 system that was prepared under micro-emulsion conditions (at high speed mixing). The product obtained under high speed mixing is compared with reactions that use conventional stirring (cf. Table 3), where greater cross-linking occurs for the former [17]. The presence of micropore sites with variable accessibility that favor THM adsorption is anticipated for intermediate levels of cross-linking, as noted for β-CD adsorbents when the inclusion site accessibility is favorable. Relatively stable inclusion complexes of THM molecules within the β-CD cavity are anticipated in water due to favorable dispersion interactions and hydrophobic effects, as outlined in a recent review for these types of inclusion complexes [22]. The role of entropy-driven processes due to hydration effects for such host–guest complexes are known to stabilize complexes between β-CD and haloalkanes [22,35,36]. Wilson and Verrall [37] reported a thermodynamic study describing the formation of complexes between β-CD and its modified forms with halothane (CHBrCl-CF_3_) in aqueous media. Moderately stable 1:1 host-guest binding constants (ca. 10^2^ M^-1^) were reported on the basis of ^19^F-NMR chemical shift variations and positive changes in the partial molar volume in aqueous media. The formation of a “facial complex” for the β-CD/halothane system provides insight on the role of polymer adsorbents that contain β-CD tend to form stable complexes with THMs. The trends in Figure 3 for such systems can be understood based on the role of inclusion binding since it correlates with the relative accessibility of the β-CD cavity interior, as described elsewhere [38,39]. The reduced sorption of the THMs in Figure 3D,E can be related to the reduced inclusion binding of such β-CD/PAA copolymers. Bromoform displays the greatest Q_m_ value overall among the THMs with the β-CD/PAA copolymers, in agreement with its greater apolar character and the key role of hydrophobic effects for adsorption by the β-CD inclusion sites. The relative ordering of the Q_m_ values for the β-CD/PAA-THMs are listed in descending order: bromoform > BDCM > DBCM > chloroform. The trend parallels the apolar character and molecular polarizabilities of the THMs (cf. Table 5). In general, hydrophobic effects provide the driving force for the more apolar (Br-containing) THMs with the β-CD containing adsorbents. For the case of SCl-1 and SCl-5 polymers (cf. Figure 3F,G), CHCl_3_ showed greater uptake over CHBr_3_ and this may relate to the presence of polar ester linkages within the polymer network. The parallel trend in uptake of THMs for SCl-5 and SCl-10 with chloroform and the similar sorption capacities for the brominated THMs concur with the role of both β-CD and cross-linker adsorption sites, where the ester linkage sites are inferred to favor dipolar interactions with CHCl_3_ [22,36,37,40].

A general comparison of the uptake profile for activated carbon (AC) with THMs revealed that the uptake was comparable to the β-CD-based copolymers (cf. Figure 3D–G). The role of hydrophobic effects for AC with the THMs was evidenced by the greater uptake of CHBr_3_, while the lowest uptake occurred for CHCl_3_, in line with the Henry’s law constants listed in Table 1. The greater uptake observed for β-CD/PAA 1:5 in Figure 3C relates to the role of macrocyclic host binding sites within the polymer compared to less preorganized (two-dimensional) graphenic surface sites of AC that reveal lower uptake of THMs in Figure 3H.

A comparison of the adsorption capacity for AC with the PS adsorbents is summarized in Table 5. The relevance of comparing the uptake results for AC with the copolymers relate to the widespread use of AC as an industrial carbonaceous adsorbent for THMs removal [3,11,12,13,15,16]. The adsorption isotherms of AC (cf. Figure 3H) are distinctive for chloroform and brominated THMs, based on differences in their polarity and molecular polarizability (cf. Table 5), as noted above. The adsorption isotherm of the AC-CHCl_3_ system displayed a Langmuir-type adsorption profile, according to the Sips (n_s_ ≈ 1) model. The greater uptake of brominated THMs by AC was assigned to the contribution of hydrophobic effects for such bromine-based THMs, based on the graphenic nature of the AC surface [41]. Several THMs display isotherm saturation at comparable Q_e_ values, further supporting the nonspecific van der Waals interactions with the AC surface and hydrophobic effects, which are prominent for the adsorption of THMs [31,35]. This is supported by the notable adsorption of CHCl_3_ by AC (Q_m_ = 0.261 mmol/g) that compares favorably with the Q_m_ values (mmol/g) for SCl-5 (0.287) and SCl-10 (0.248). In contrast, AC had lower adsorption of CHBr_3_ (Q_m_ = 0.141) compared with β-CD/PAA 1:5 (Q_m_ = 1.07). The relative adsorption capacity (Q_m_; mmol/g) of the PS material with CHCl_3_ at similar experimental conditions showed the following trend: SCl-5 (0.287) > AC (0.261) > SCl-10 (0.248) > CP-5 (0.141) > β-CD/PAA 1:10 (0.111) > β-CD/PAA 1:5 (1.104) > β-CD/PAA 1:5 at high speed (0.0485) ~ CP-1 (0.0420).

Table 4 and Table 5 illustrate that variable Q_m_ values are noted for the copolymers and the AC materials for the various THMs. The carbonaceous materials cover a range of systems such as powdered AC (PAC), carbon nanotubes (CNTs), and various AC fibers (AC-10, AC-15, and AC-20) [42,43]. The Sips exponent parameter (n_s_) for the copolymers deviates from unity (i.e., n_s_ ≠ 1) and indicates the presence of multiple adsorption sites. For CD- and CS-based copolymers, the availability of multiple adsorption sites depend largely on the structure and composition of the PS materials (cf. Table 3). There are two potential adsorption sites for β-CD polymers that are known due that arise from the β-CD inclusion sites and the interstitial (non-inclusion) sites, where the latter are assigned the cross-linker sites [44,45]. The role of dual adsorption sites in CD-based copolymers is strongly supported by dye-adsorption [17,18,19] and mass spectrometry studies [44]. It follows that the role of multiple adsorption sites for copolymers of CS and PAA relate to the presence of micropore domains due to cross-linking effects and surface bound groups of the polymer framework [19]. The latter includes hydrogen bond donor–acceptor sites (e.g., –NH, –OH, and –COO^−^) that likely contribute to such secondary adsorption sites for cross-linked copolymers [46]. The relative adsorption capacities of the carbonaceous adsorbents (cf. Table 4) and the copolymers (cf. Table 3) with the THMs in water highlight the unique adsorption properties of such PS-based materials. Taken together with the sustainability and biodegradability of PS-based adsorbents, the above results indicate their significant potential for targeted contaminant removal in advanced water treatment processes [11,13].

## 3. Materials and Methods

### 3.1. Reagents

Methanol (HPLC grade) was obtained from Fisher Scientific. Activated carbon (AC, Darco 20 × 40 LI) was purchased from Norit America (Boston, MA., USA) and was treated with 2.0 M HCl using a Soxhlet extractor for 48 h. All water used in this work was distilled and deionized water. All the solutions were freshly made prior to analysis.

#### 3.1.1. Copolymer Polysaccharide Materials

Copolymers that contain polysaccharides such as β-CD and chitosan (β-CD/PAA 1:5, β-CD/PAA 1:5 at high speed, β-CD/PAA 1:10, β-CD/SCl 1:5 and 1:10) and chitosan/PAA copolymers (CP-1 and CP-5) are summarized in Table 3. A detailed account of the synthesis and materials characterization are outlined elsewhere [17,18,19].

#### 3.1.2. Internal Standard

The internal standard for GC analysis was a 1% (*v*/*v*) chlorobenzene solution (Sigma-Aldrich Canada; Oakville, Ontario) in methanol.

#### 3.1.3. Stock Standard Solutions

The THMs calibration standard is a mixture of chloroform, BDCM, DBCM, and bromoform obtained from ULTRA Scientific (Santa Clara, CA., USA) at 5000 ppm (*w*/*v*) in methanol. 

### 3.2. Apparatus

Custom built (9 mL) glass vials with silicone- and Teflon-lined caps were designed in-house for the sorption experiments to achieve an efficient gas-tight seal suitable for the analysis of volatile organic carbons (VOCs). A 5890 Hewlett Packard model gas chromatograph (Wilmington, DE, USA) equipped with an electron-capture detector (ECD) and a cool on-column (DV-1, 30 m × 0.32 mm ID × 25 μm) injection system.

### 3.3. Operating Conditions

A helium carrier gas was maintained at a constant 10-kPa inlet pressure. The makeup gas was 5% (*v*/*v*) methane–argon mixture with a flow rate of 60.0 mL/min. The injector and detector temperatures were set at 250 °C. The maximum oven temperature was 300 °C and the equilibrium time was set at 3.00 min. For GC analysis of the THMs, the oven temperature and initial temperature (50 °C), and the final temperature was 100 °C. The initial time was 5.00 min. The heating rate was 10.0 °C/min and the final time was 2.50 min.

### 3.4. Methods

The DAI method [10] was used for the GC analysis without extraction or any pre-concentration steps. The aqueous solutions of THMs were directly injected to the gas chromatograph using 1 μL of analyte with 1 μL of air into the injector port for every analysis and analyzed in triplicate.

#### 3.4.1. Calibration of Standard Solutions.

The standard solution of THMs was diluted to 50 ppm. Eight data points ranging from 5–50 ppm were collected to construct a quantitative peak area vs. concentration calibration curve. The calibration curves were plotted by concentration versus area ratios, according to Equation (1).
(1)$$area\;ratio=\;\frac{A_{S}}{A_{I}}$$
where ‘A_S_’ is the relative peak area of the respective component THMs and ‘A_I_’ is the peak area of the internal standard.

#### 3.4.2. Adsorption Isotherms

Fixed amounts (~5–6 mg) of the powdered copolymer materials were sieved through size 40-mesh sieves (particle size < 420 μm) and mixed with 7 mL of blank solution at variable concentration (5–50 ppm) for each of the component THMs until fully equilibrated on a horizontal shaker table for 24 h. A 4% (*v*/*v*) internal standard was added 15 min. ahead of each sample injection of the supernatant solution from the sample vial containing copolymer solutions after completed gravity settling of the solution phase. The data corresponding to the isotherms were analyzed in triplicate.

The adsorption isotherms are presented as plots of the adsorbed amount of THMs in the copolymer phase per mass of adsorbent (Q_e_; mmol/g) versus the equilibrium residual concentration of unbound THMs (i.e., adsorbate) in aqueous solution (C_e_) after the sorption process. Q_e_ was estimated by Equation (2) where C_0_ and Ce are the concentrations of THMs before and after the sorption process, respectively. V is the volume (L) of solution and m is the mass (g) of adsorbent.
(2)$$Q_{e}=\frac{\left(C_{0}-C_{e}\right)\times V}{m}$$

The Sips isotherm model [34] (cf. Equation (3)) is a versatile and generalized isotherm that accounts for a distribution of adsorption energies on the sorbent surface with behavior that may vary from a monolayer (Langmuir) to multi-layer (Freundlich) type profile. The parameter (n_s_) reflects the heterogeneity of the sorbent where a value of n_s_ = 1 infers homogenous surface, whereas n_s_ ≠ 1 indicates heterogeneous surface sites. Langmuir-type behavior is predicted when n_s_ = one, and Freundlich behavior occurs when the exponent term in the denominator of Equation (3) is much less than one.
(3)$$Q_{e}=\frac{Q_{m}K_{s}C_{e}{}^{n_{s}}}{1+K_{s}C_{e}{}^{n_{s}}}$$

*K_s_* represents the Sips equilibrium constant; *Q_m_* is the maximum monolayer adsorption capacity per unit mass of sorbent; and *n_s_* is an exponent parameter, as described above.

The criteria of the “best-fit” between the calculated isotherm and the experimental data are determined by the correlation coefficient (R^2^) and the chi-square distribution (χ^2^). The parameter R^2^ ≈1 denotes a satisfactory “goodness-of-fit”. However, a more sensitive measure for non-linear least squares fitting involves the minimization of the χ^2^ parameter. The chi-squared distribution is commonly used to test the “goodness-of-fit” between an observed and a theoretical distribution. The independence of the two criteria of classification of qualitative data, and in the confidence interval estimation for a population standard deviation of a normal distribution derived from a sample standard deviation. χ^2^ is defined by Equation (4) according to the chi-square distribution with *k* degrees of freedom.
(4)$$\chi^{2}=\sum_{i=1}^{k}\frac{{\left(Q_{o,i}{-Q}_{e,i}\right)}^{2}}{Q_{e,i}}$$

Q_o,i_ represents the observed value and Q_e,i_ is the theoretical value.

#### 3.4.3. Error Analysis

The calculation of uncertainty for arithmetic operations of several estimates associated with measurements, each of which has a random error, is not solely the sum of individual errors. The relationship between the concentration terms (cf. Equation (2)) from instrumental estimates are based on multiplication and division. Therefore, Equation (5) was used for the error analysis [47].
(5)$$\%e_{4}=\sqrt{{\left(\%e_{1}\right)}^{2}+{\left(\%e_{2}\right)}^{2}+{\left(\%e_{3}\right)}^{2}}$$

In the case of the error calculation for Q_e_, the uncertainties are attributed as follows: %e_1_ is estimated from calibration curves under the similar experimental conditions; and %e_2_ is estimated from the detection limit of analytical balance (≈10^−5^ g) for weight measurements. The term, %e_3_, was estimated from the detection limit volume (0.06 mL) of 10 mL Kimax-51 petite tubes (Sigma-Aldrich Canada, Oakville, ON.) to obtain the volume of solution. The overall error (%e_4_) was implemented into the fitting program, Origin 7.5 for the estimation of the total error.

## 4. Conclusions

In this work, a systematic adsorption study of a diverse range of polysaccharide-based polymers with THMs spiked into water samples were reported and compared with carbonaceous adsorbents (cf. Table 4 and Table 5). The level of THMs in aqueous solution at pH 6.5 and 295 K was evaluated using GC-ECD with a direct analysis injection method. The polymers that contained either β-CD or chitosan (CS) displayed variable uptake of THMs where the adsorption of apolar THMs appear to be governed by hydrophobic effects, along with secondary dipolar interactions with the polar surface functional groups of the copolymer. Polymers that contain β-CD displayed highly favorable uptake of THMs due to the formation of inclusion complexes to the overall adsorption process. The role of hydrophobic effects and inclusion complexes is noted by the favorable uptake of CHBr_3_ by β-CD/PAA (1:5) and CS/PAA (1:1). The Sips model provided estimates of the monolayer sorption capacity (Q_m_; mmol/g) for the various copolymers, as follows: CHCl_3_ (0.0485–0.287); DBCM (0.0712–0.277); BDCM (0.0684–0.387); and CHCl_3_ (0.0522–1.07). The copolymers displayed evidence of molecular selective affinity among the various component THMs (cf. Figure 3 and Table 4). The structural role of polysaccharide and cross-linker units contribute uniquely to variable physiochemical properties of the adsorbent material, according to the synthetic conditions (cf. Table 3). The biodegradable adsorbents reported herein displayed tunable adsorption properties that rival those of commercial activated carbon adsorbents. This study demonstrates that variable cross-linker content and judicious choice of the polysaccharide afford the design of adsorbent materials with improved properties for the controlled removal of waterborne THMs. Further research is underway to evaluate the kinetic properties of such polysaccharide adsorbents with real environmental water samples that contain THMs.

## Data Availability

The data presented in this study are available on request from the corresponding author.

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
