# Peer review of "Adsorption Studies of Waterborne Trihalomethanes Using Modified Polysaccharide Adsorbents"

_molecules, 2021, doi:10.3390/molecules26051431_

Round 1

Reviewer 1 Report

In the manuscript titled " Adsorption Studies of Waterborne Trihalomethanes Using Modified Polysaccharide Adsorbents", the authors present the study on adsorption properties of different polysaccharide based adsorbents copolymer adsorbents for waterborn contaminants. Over all the the study is interesting.

There are  a few comments, questions and suggestions for the authors.

1) The authors used the term ambient pH in the manuscript, What is the pH of ambient pH? Do the authors mean neutral pH?

2) Long result and discussion sections without sub section is very hard to follow. Should be divided into multiple subsections

3) Author only present sorption isotherms for copolymer adsorbents and for pure ones as control for e.g. (Just beta CD, and PAA). The results on control only should be provided to justify the use of copolymer adsorbents

4) In the material and methods section, the authors give  only the short form of the adsorbents and refer to 3 different references to know what those short forms mean. At least authors should provide a summary of table of the samples used and their important properties for e.g. what does beta-CD/PAA 1:5 at high speed mean. Its confusing and for a reader to go back to references to understand further, is kind of confusing.

Author Response

Response to Reviewer comments on Manuscript ID:  molecules-1112555

Reviewer #1

In the manuscript titled " Adsorption Studies of Waterborne Trihalomethanes Using Modified Polysaccharide Adsorbents", the authors present the study on adsorption properties of different polysaccharide based adsorbents copolymer adsorbents for waterborn contaminants. Over all the the study is interesting.

There are  a few comments, questions and suggestions for the authors.

  • The authors used the term ambient pH in the manuscript, What is the pH of ambient pH? Do the authors mean neutral pH?

Response:  The explicit pH was specified throughout the manuscript (pH 6.5), as noted in the updated version.

  • Long result and discussion sections without sub section is very hard to follow. Should be divided into multiple subsections

Response:  The use of subsections was implemented as suggested by the reviewer.

Author only present sorption isotherms for copolymer adsorbents and for pure ones as control for e.g. (Just beta CD, and PAA). The results on control only should be provided to justify the use of copolymer adsorbents

Response:  The use of PAA and CD are not amenable as controls since these polysaccharides are water soluble and the adsorption method herein relies on an insoluble adsorbent to ensure phase separation of the unbound adsorbate and the adsorbent (cf. eqns (2) and (3)).  The role of CD and its crosslinked forms have been evaluated by a dye adsorption assay (see Refs. [1-3] below). Cross-linking serves to alter the accessibility of the inclusion sites of b-CD, whereas greater cross-linking attenuates the formation of inclusion complexes due to steric effect. The presence of cross-linkers in copolymers of the type reported herein may also serve as secondary binding sites; thus, it is misleading to use either CD or PAA as control systems. As well, the formation of copolymers alters the hydrophile-lipophile balance (HLB) of the system. Marked changes in HLB are very likely in the case of b-CD/PAA and Chitosan/PAA copolymers due to the reduction in –COOH groups upon formation of ester and amide linkages and relative to the precursors (CD, PAA, etc.), in agreement with the results of Ref. [2] given below.

References

[1]            Mohamed, et al.  Estimation of Surface Accessibility of b-Cyclodextrin in Cyclodextrin-Copolymer Materials, Carbohydr. Polym., 2010, 80, 186-196.

[2]            Mohamed, et al., J. V.  Evaluation of the Accessible Inclusion Sites in Copolymer Materials containing β-Cyclodextrin,  Carbohydrate Polymers, 2012, 87, 1241-1248.

[3] Mohamed, et al.  Thermodynamic Properties of Inclusion Complexes Between β-Cyclodextrin and Naphthenic Acid Fraction Components, Energy and Fuels, 2015, 29 (6), 3591–3600.

4) In the material and methods section, the authors give only the short form of the adsorbents and refer to 3 different references to know what those short forms mean. At least authors should provide a summary of table of the samples used and their important properties for e.g. what does beta-CD/PAA 1:5 at high speed mean. Its confusing and for a reader to go back to references to understand further, is kind of confusing.

Response:  The use of a Table that summarizes the acronym and compositions of the copolymer materials was incorporated into the revised manuscript (see Table 3).

The authors wish to acknowledge Reviewer #1 for the insightful and constructive comments, along with the opportunity to improve its overall quality. We have further edited the manuscript for clarity, language, and syntax throughout to meet the high standards of this journal.

Reviewer 2 Report

The paper of Guo, Bharadwaj and Wilson (molecules-1112555) presents an study of trihalomethanes adsorption by modified polysaccharides.

The paper contain equilibrium isotherms for four trihalomethanes onto seven different adsorbents  (2 modified chitosan and  5 modified ciclodextrines), this implies a great amount of experimental work. Discussion is properly done taking into account the effect of the cross-linker in the adsorption capacity of the polysaccharide and the physicochemical properties of both trihalomethanes and adsorbents. Results show high adsorption capacities, even higher than active carbon so these materials can be useful for water decontamination.

So I can recommend publication after minor revision.

Some minor comments are the following:

-CD/PAA 1: 5 presents the higher trihalomethanes adsorption capacity (Fig. 3-C and table 3). In particular, Qm value for bromoform is unusual high. This behavior is due to the fact that the experimental data do not reach the plateau. Therefore, fitting to Sips model does not give good results. Better results can be obtained including more experimental points at higher concentrations.

-The use of the Sips model, instead of Langmuir, involves including a parameter (n) associated with the heterogeneity of the material. There is a lack of discussion about the values of this parameter. For example, for the same material (SCl-10), n varies from 0,7 to 4.4. Is there any explanation?

-Discussion at the end of page 6: “A comparison of the 1:1 and 1:5 chitosan-PAA polymer (CP-1 and CP-5) reveals greater uptake of chloroform for CP-1 over other THMs, according to the following trend in uptake: CHCl3 > BDCM > DBCM > CHBr3. By contrast CP-5 shows greater uptake of bromoform…”     But the behavior is just the opposite (see fig 3A-B and table 3)

-Introduction, pag 2: are you sure that reference [10] (Clesceri, Greenberg…) is the proper reference at this point?

-Introduction, pag. 1. :”…where X represents fluorine, chlorine bromine or iodine (cf. Scheme 1)”. But scheme 1 is about structure of polysaccharides.

-table 3 and 4: superscripts a and b, are not explained in the legend

-section 4.1.1: what is the meaning of superscripts 13, 14 and 15?

Author Response

Response to Reviewer comments on Manuscript ID:  molecules-1112555

Reviewer #2

The paper of Guo, Bharadwaj and Wilson (molecules-1112555) presents an study of trihalomethanes adsorption by modified polysaccharides.

The paper contain equilibrium isotherms for four trihalomethanes onto seven different adsorbents  (2 modified chitosan and  5 modified ciclodextrines), this implies a great amount of experimental work. Discussion is properly done taking into account the effect of the cross-linker in the adsorption capacity of the polysaccharide and the physicochemical properties of both trihalomethanes and adsorbents. Results show high adsorption capacities, even higher than active carbon so these materials can be useful for water decontamination.

So I can recommend publication after minor revision.

Some minor comments are the following:

-CD/PAA 1: 5 presents the higher trihalomethanes adsorption capacity (Fig. 3-C and table 3). In particular, Qm value for bromoform is unusual high. This behavior is due to the fact that the experimental data do not reach the plateau. Therefore, fitting to Sips model does not give good results. Better results can be obtained including more experimental points at higher concentrations.

Response:  We agree with the reviewer that the isotherms have not reached the plateau as indicated. A key aspect of the study was the use of equimolar solutions in the case of mixed THMs to enable relative evaluation of the THMs via GC analysis. The notable uptake is attributed to the favourable uptake of bromoform by b-CD due to its apolar character and the nature of the copolymer material. Additional interpretation and revision of the results/discussion is provided in the revised manuscript.

-The use of the Sips model, instead of Langmuir, involves including a parameter (n) associated with the heterogeneity of the material. There is a lack of discussion about the values of this parameter. For example, for the same material (SCl-10), n varies from 0,7 to 4.4. Is there any explanation?

Response:  The use of the Sips isotherm provides an account of Langmuir and Freundlich isotherm behavior since it is a hybrid model of both models. In turn, the exponential (ns) term of the Sips model provides an indication of the heterogeneity of the surface sites, especially when ns deviates from unity. For the case when where ns = 1, the Sips model converges to that of the Langmuir model, where it accounts for adsorption onto homogeneous adsorption sites.  The variation of ns provides support for multiple adsorption sites when the exponential term ns¹ 1. The latter occurs when adsorption occurs at two or more sites, and concurs with the expectation for copolymers with multiple binding sites that participate in the adsorption process. This has been described in some detail in the following reference by Mohamed et al. (see Energy and Fuels, 2015, 29 (6), 3591–3600).  An account of the above was incorporated into the revised manuscript.

-Discussion at the end of page 6: “A comparison of the 1:1 and 1:5 chitosan-PAA polymer (CP-1 and CP-5) reveals greater uptake of chloroform for CP-1 over other THMs, according to the following trend in uptake: CHCl3 > BDCM > DBCM > CHBr3. By contrast CP-5 shows greater uptake of bromoform…”     But the behavior is just the opposite (see fig 3A-B and table 3)

Response:  The inconsistencies were addressed in the revised version of the manuscript.

-Introduction, pag 2: are you sure that reference [10] (Clesceri, Greenberg…) is the proper reference at this point?

Response:  This reference was updated in the revised version.

-Introduction, pag. 1. :”…where X represents fluorine, chlorine bromine or iodine (cf. Scheme 1)”. But scheme 1 is about structure of polysaccharides.

Response: The appropriate corrections were applied.

-table 3 and 4: superscripts a and b, are not explained in the legend

-section 4.1.1: what is the meaning of superscripts 13, 14 and 15?

Response:  The items above have been properly defined and corrected in the revised manuscript.

The authors wish to acknowledge Reviewer #2 for the insightful and constructive comments, along with the opportunity to improve its overall quality. We have further edited the manuscript for clarity, language, and syntax throughout to meet the high standards of this journal.

Reviewer 3 Report

 Generally, the manuscript is interesting and worth considering for publication. However, there are some points which should be clarified or improved

  1. Please, add information about the sorbents, especially the physical properties (grain size, bulk density, and so on…)
  2. Please added the city and country to the instruments using in the studies.
  3. Maybe it should be studied the dosage of adsorbent materials and the influence of other parameters (pH, contact time, temperature). For example: at higher adsorbent dosage, at another pH, contact time and temperature, high absorption capacity can be obtained.
  4. I think the Langmuir and Freundlich equilibrium isotherms also should be introduced in this study. How many replicates were of isotherm studies?
  5. More detail is needed here on the mechanism of interactions that lead to the observed performance of adsorbent materials. The mechanism behind the adsorption has to be studied in detail.
  6. Adsorption and desorption studies of the adsorbent materials should be performed.

Author Response

Response to Reviewer comments on Manuscript ID:  molecules-1112555

Reviewer #3

Generally, the manuscript is interesting and worth considering for publication. However, there are some points which should be clarified or improved

  1. Please, add information about the sorbents, especially the physical properties (grain size, bulk density, and so on…)

Response:  Additional information was provided in Table 3 and in the experimental section relevant to this adsorption study in the revised manuscript.

  1. Please added the city and country to the instruments using in the studies.

Response:  This information will be added in the revised Manuscript.

  1. Maybe it should be studied the dosage of adsorbent materials and the influence of other parameters (pH, contact time, temperature). For example: at higher adsorbent dosage, at another pH, contact time and temperature, high absorption capacity can be obtained.

Response:  Due to the relatively low solubility of these THMs, according to the Henry’s law constants, it was not practical to carry out the suggested measurements. Therefore care was taken to ensure that the systems were at equilibrium in order to obtain reliable concentrations of free THMs in water to be reliably measured by the GC-DAI method [see ref. 10] reported herein.

  1. I think the Langmuir and Freundlich equilibrium isotherms also should be introduced in this study. How many replicates were of isotherm studies?

Response:  A discussion of the isotherms along with relevant citations are provided. The data in the isotherms were obtained as triplicate measurements, as outlined in the revised manuscript.

  1. More detail is needed here on the mechanism of interactions that lead to the observed performance of adsorbent materials. The mechanism behind the adsorption has to be studied in detail.

Response:  We agree that the mechanism of the adsorption process should be studied in greater detail. This is beyond the scope of the present study based on our objectives and is planned as part of future work as noted in the conclusion section.  The reviewer is directed to the first comprehensive thermodynamic and structural study for a related halogenated alkane (halothane), as outlined in the following study: https://doi.org/10.1139/cjc-2014-0549

As such, a more detailed follow-up study is planned as part of future work to extend beyond the objectives of the present work.

  1. Adsorption and desorption studies of the adsorbent materials should be performed.

Response:  While we agree that desorption studies may be of potential interest, it is well known that the stability of free THMs in aqueous solution are governed by Henry’s law. As such, the THMs can be outgased from aqueous solution by several strategies:

  1. reducing the vapour pressure by venting the solutions to the open atmosphere
  2. bubbling a secondary gas into the solution such as nitrogen to reduce the partial pressure of the THMs in the system
  • heating the solution since the solubility of THMs in water decrease markedly with increasing temperature.

In relation to the above, the role of complex stability for b-CD/halothane complexes was established in the reference below due to the need to prepare samples in sealed glass ampoules, as described in the following study:   https://doi.org/10.1139/cjc-2014-0549

The current study extends that knowledge by the use of custom made glass vessels with sure-tight seals and Teflon liner caps. In the absence of gas-tight seals, it was not possible to achieve adequate equilibrium concentrations, in agreement with the Henry’s law constants, as presented in Table 1.

In summary, the authors wish to acknowledge Reviewer #3 for the insightful and constructive comments, along with the opportunity to improve its overall quality. We have further edited the manuscript for clarity, language, and syntax throughout to meet the high standards of this journal.

Round 2

Reviewer 1 Report

The authors' responses and the revision of the manuscript are to the sufficient degree.

Reviewer 3 Report

I am satisfied with the revision. The paper meets the necessary standards for publication in  Molecules Journal.